# Positively Charged Gold Quantum Dots: An Nanozymatic “Off-On” Sensor for Thiocyanate Detection

**DOI:** 10.3390/foods11091189

**Published:** 2022-04-19

**Authors:** Syed Rahin Ahmed, Masoomeh Sherazee, Seshasai Srinivasan, Amin Reza Rajabzadeh

**Affiliations:** School of Engineering Practice and Technology, McMaster University, 1280 Main Street West Hamilton, Hamilton, ON L8S 4L8, Canada; ahmes91@mcmaster.ca (S.R.A.); sherazem@mcmaster.ca (M.S.); ssriniv@mcmaster.ca (S.S.)

**Keywords:** gold quantum dots, nanozyme, colorimetric detection, thiocyanate detection

## Abstract

The concentration of thiocyanate (SCN^−^) in bodily fluids is a good indicator of potential and severe health issues such as nasal bleeding, goiters, vertigo, unconsciousness, several inflammatory diseases, and cystic fibrosis. Herein, a visual SCN^−^ sensing method has been developed using the enzyme-like nature of positively charged gold quantum dots (Au QDs) mixed with 3,3′,5,5′-*tetramethylbenzidine* (TMB) and hydrogen peroxide (H_2_O_2)_. This research also reports a new method of synthesizing positively charged Au QDs directly from gold nanoparticles through a hydrothermal process. Microscopic imaging has showed that the Au QDs were 3–5 nm in size, and the emission wavelength was at 438 nm. Au QDs did not display any enzyme-like nature while mixed up with TMB and H_2_O_2_. However, the nanozymatic activity of Au QDs appeared when SCN^−^ was included, leading to a very low detection limit (LOD) of 8 nM and 99–105% recovery in complex media. The steady-state kinetic reaction of Au QDs showed that Au QDs had a lower Michaelis–Menten constant (*Km*) toward H_2_O_2_ and TMB, which indicates that the Au QDs had a higher affinity for H_2_O_2_ and TMB than horseradish peroxidase (HRP). A mechanism study has revealed that the scavenging ability of hydroxyl (•OH) radicals by the SCN^−^ group plays an important role in enhancing the sensitivity in this study. The proposed nanozymatic “Off–On” SCN^−^ sensor was also successfully validated in commercial milk samples.

## 1. Introduction

Thiocyanate (SCN^−^) is a well-known inhibitor of the sodium-iodide symporter, which is responsible for concurrently transporting sodium and iodide ions from extracellular fluids into thyroid epithelial cells. Upon over-inhibition of this system, the regulation of the thyroid gland becomes inefficient, and depleted levels of the appropriate ions lead to health issues [1,2]. In tandem, low concentrations of SCN^−^ promote the efflux of iodide and reduce the proper amount of iodide [3]. SCN^−^ is organically present in urine, saliva and blood. It also serves as an antioxidant in the immune system, working to protect the lungs from overexposure to hydrogen peroxide (H_2_O_2_) and suppressing myeloperoxidase from triggering health complications [4]. In excess, SCN^−^ is dangerous, as it can cause iodine deficiency, which can subsequently lead to more severe health issues such as nasal bleeding, goiters, vertigo, and unconsciousness [5]. Antagonistically, the absence of SCN^−^ also causes health problems, such as several inflammatory diseases and cystic fibrosis [6].

There are numerous sources of SCN^−^; the most common source, for human concern, would be food. There are several widely available foods, such as broccoli, kale, cabbage, lima beans, and sweet potatoes, that contain either the SCN^−^ anion itself or its metabolic precursors and/or by-products. Moreover, it is naturally present in dairy milk in lesser concentrations, and is often added in surplus because of its bacteriostatic effects of increasing the shelf-life of the dairy product [7]. As a result, it is necessary to monitor the level of SCN^−^ in the human body, as well as in sites and containers of household resources intended for human consumption, namely, commercial food crops, wastewater systems, food processing plants, and various biological samples. Numerous SCN^−^ detection methods have been proposed over the last few years; for example, traditional colorimetric assays, the chromatographic method, the fluorescence method, the electrophoresis method, the chromatography technique, the electrochemical method and the SERS technique [8,9,10,11]. Unfortunately, these techniques need a longer time, more chemicals, and are costly. Hence, a sensitive, reliable, and cheaper SCN^−^ detection technique is required.

The integration of nanomaterials into food science has become attractive and revolutionized various scientific and industrial fields. Particularly, the biosensing application of nanomaterials in the food sector, achieved through the colorimetric method, the electrochemical method, the fluorescence method or the surface-enhanced Raman scattering technique, has gained much research interest because of its effortless preparation, stability, bioconjugation and improved sensitivity [12]. Among different techniques, the nanomaterials-assisted colorimetric detection method has received much attention, as it offers low-cost portable sensing devices and visual quantification of target analytes. However, besides its several advantages, the low sensitivity of the colorimetric detection technique is a significant drawback that raises a big concern as to its real-life applications. Over the last few years, the enzyme-like nature of nanomaterials (nanozymes) has been used to overcome the drawbacks of traditional colorimetric methods, for example, the surface plasmon resonance-based colorimetric detection method. Nanozymes are inorganic enzymes with several advantages over their organic counterparts, derived from their adaptability, robustness under miscellaneous environmental conditions, easy mass-producibility, relatively low cost, higher stability, and long-term storage under optimal conditions. Furthermore, compared to other artificial enzymes, nanozymes are inherently superior due to their structure (i.e., shape, size, and composition). This allows them to perform multifunctional catalysis, undergo modification, have programmable responses to external stimuli, self-assemble, and be deconstructed [13]. With such coveted characteristics, nanozyme applications are vast and varied, ranging from the likes of therapeutics to biosensing and even environmental protection [14,15,16,17,18]. To date, a large number of nanoparticles, their composites, and quantum dots have shown unique nanozymatic activity. Among the different nanomaterials, the nanozymatic activity of gold nanoparticles (Au NPs) is popular because of their many benefits, including a higher absorption coefficient, non-cytotoxicity, and their large surface area, providing ample real estate for modification, carrying, and reaction [19,20,21,22,23,24,25,26,27]. Moreover, in biological settings, Au NPs have shown better compatibility, bioconjugation abilities and tunable properties. Additionally, because of their size, shape, and crystalline structure, Au NPs have been easily implemented as therapeutic agents and vaccine carriers, since they can swiftly travel into target cells carrying comparatively higher drug quantities [28,29]. It is well-known that positively charged Au NPs show better nanozymatic activity because of their strong electrostatic interaction with •OH radicals. Moreover, the smaller-sized nanomaterials achieved better nanozymatic activity compared to their bulky counterparts because of their larger surface area, which allows more substrate to interact with it. Considering all these parameters, the synthesis of positively charged Au QDs might introduce a new era in nanozymatic biosensing applications. Hence, the present study aims to develop a new synthesis method of positively charged Au QDs, and furthermore, a nanozymatic “Off–On” sensing mechanism for SCN^−^ detection has been presented. At first, positively charged Au QDs were synthesized via a new chemical hydrothermal process. Upon conversion from Au NPs to Au QDs, the nanomaterials lose their initial nanozymatic activity. However, the nanozymatic property can be restored in the presence of SCN^−^, enabling one to monitor SCN^−^ by detecting the intensity of its characteristic blue color, which emanates from the nanozymatic reaction.

## 2. Materials and Methods

### 2.1. Material

Chloroauric acid (HAuCl_4_. 3H_2_O), sucrose, sodium thiocyanide, PBS buffer (7.5), BSA, fructose, lysine, CaCl_2_, TMB, Na_2_SO_4_, H_2_O_2_, Na_2_CO_3_, cysteamine, EDTA, sodium borohydride (NaBH_4_), NaCl, CH_3_COONa, MgCl_2_, and NaNO_3_ were purchased from Sigma-Aldrich, Oakville, Ontario, Canada.

### 2.2. Preparartion of Positively Charged Au NPs

The positively charged Au NPs were synthesized using the previously reported procedure [30]. For example, 213 mM of cysteamine solution (400 μL) was mixed with 1.42 mM HAuCl_4_ solution (40 mL) and stirred for 20 min. Then, NaBH_4_ solution (10 μL, 10 mM) was added to it, and stirring was continued for an additional 30 min. After purification, the synthesized Au NPs were stored at 4 °C.

### 2.3. Preparation of Positively Charged Au QDs

The positively charged Au QDs were obtained through a hydrothermal method without any additional chemicals. For example, 15 mL of freshly prepared positively charged Au NPs were heated in an oven (200 °C, 2 h) using an autoclave tube. The red color Au NPs solution turned yellow, indicating the formation of Au QDs.

### 2.4. The Sensing Protocol of SCN^−^

In this study, the detection of SCN^−^ was performed as follows: first, different concentrated solutions of SCN^−^ (50 µL) were prepared in PBS (7.5) buffer and mixed with the Au QDs solution (50 µL, 0.01 mg mL^−1^) separately. Then, the mixture was gently heated at 100 °C for 10 min. After cooling down to room temperature, a 5 mM TMB and 10 mM H_2_O_2_ mixture solution (50 µL) was added into each sample, and the color turned to blue. The color intensity was monitored using a UV-visible spectrometer (Synergy H1, BioTek, Winooski, VT, USA).

### 2.5. Terephthalic Acid (TA) Experiment

The Terephthalic Acid test was accomplished to check the production of •OH due to the nanozymatic reaction [30]. It is well-established that in the presence of TA, H_2_O_2_ molecules convert to •OH radicals and generate a fluorescence signal. Here, the solution of TA (0.5 mM, 50 μL) was added with SCN^−^-capped Au QDs (50 μL) for 30 min in the presence of H_2_O_2_ solution. Then, the changes in fluorescence signal were registered with a spectrophotometer (Synergy H1, BioTek, Winooski, VT, USA).

## 3. Results and Discussions

### 3.1. The Detection Mechanism of SCN^−^

The schematic presentation of the proposed mechanism is illustrated in Figure 1. Two main varieties play crucial roles in this sensing method. One is the strong interaction of the thiol group with the surface of gold nanomaterials, which offers effortless and specific conjugation. The other is the ^•^OH radical scavenging activity of the SCN^−^ group, which performs a crucial role in enhancing the sensitivity of biosensors through initiating the radical chain reaction and oxidizing TMB molecules [31]. Here, the cysteamine group that is attached on the Au NPs acts as the reducing and capping agent of Au QDs. Upon the addition of SCN^−^, the surface of the Au QDs will be covered by SCN^−^ group and will stabilize through the thiol–Au interaction or disulfide bond. Next, the introduction of the H_2_O_2_ and TMB mixture solution promulgates the development of a blue color in the solution. The absorbance intensity of the blue-colored solution is proportionate to the SCN^−^ amount, and allows the quantitative measurement of SCN^−^.

### 3.2. Characterization of Au NPs and Au QDs 

As presented in Figure 2A, the absorbance spectra of the synthesized Au NPs and Au QDs appeared at 525 nm and 430 nm, respectively. The fluorescence spectroscopy study of Au NPs and Au QDs is presented in Figure 2B. As seen in this figure, the emission wavelength of the Au QDs appears at 450 nm. However, no such emission peak is observed for Au NPs. The microscopic study of Au NPs and Au QDs was performed using FEI Titan 80–300 LB (Hillsboro, OR, USA) and showed that the size was approximately 35–40 nm and 3–5 nm (Figure 2C,D), respectively. The lattice spacing of Au QDs was 0.244 nm, which is associated with the Au NPs’ *d*-spacing (111) [32] (Figure 2E). The zeta potential value of Au NPs and Au QDs was 24.1 mV and 23.9 mV, respectively. The fluorescence lifetime of Au QDs was measured using a DeltaFlex TCSPC Lifetime Fluorometer (Horiba, London, ON, Canada) and the calculated value was 2.6 ns.

### 3.3. Nanozymatic Activity of Au NPs and Au QDs

When TMB and H_2_O_2_ were added into the Au NPs solution, a clearly differentiable peak appeared at 655 nm, whereas no peak appeared for Au QDs (Figure 3A). Most probably, the nanozymatic activity of the Au QDs disappeared because of the high-temperature reaction that changed the surface properties. However, the characteristic nanozymatic peak appeared when Au QDs were heated with a differentially concentrated SCN^−^ solution, and a calibration curve was created with the absorbance value at various concentrations of SCN^−^ solution (Figure 3B). As seen in this figure, the absorbance has a linear correlation with SCN^−^ concentration and the calculated LOD was 8 nM.

### 3.4. Kinetic Study on the Nanozymatic Activity of Au QDs

An apparent steady-state kinetic study was utilized to investigate the reaction kinetic parameters of SCN^−^-capped Au QDs. At first, the non-enzymatic reaction of SCN^−^-capped Au QDs was performed at different concentrations of H_2_O_2_ and TMB with variable times. Then, the Michaelis–Menten equation (Figure 4A,B) and Lineweaver–Burk double reciprocal plot (Figure 4C,D) were employed to get the kinetic reaction parameters. The results of the kinetic study have revealed that the Au QDs had a lower *K*m value (0.38) towards TMB compared to HRP (0.43). Moreover, the as-synthesized Au QDs had a lower *K*m value (0.34) towards H_2_O_2_ than HRP (3.7). These results indicate the stronger affinity of Au QDs towards H_2_O_2_ and TMB [20]. The catalytic rate constant (*K*cat) of Au QDs (5.80 × 10^3^ s^−1^) was higher than HRP (4.00 × 10^3^ s^−1^), indicating the non-enzymatic reaction of Au QDs was kinetically faster than the natural enzyme (HRP) [33].

There are two features that strongly enhance the color of the reaction and the assay sensitivity in this study. First, the small size of the nanomaterials helps enhance the enzyme-like activity by providing a large surface area that aids in the attachment of a higher amount of the substrate. Second, the SCN^−^ is prominent for its •OH scavenging ability, which improves the enzyme-like nature [34]. Here, the enzyme-like activity of Au QDs starts in the company of TMB and H_2_O_2_ (Figure 5). In this, initially, H_2_O_2_ decomposes to •OH radicals, and they attach onto the surface of the Au QDs (Figure 5i). The scavenging activity of SCN^−^ helps to hold the radicals and facilitates the TMB oxidation (Figure 5ii). Finally, a deep blue-colored solution will develop due to the catalytic oxidation of TMB, which can be linearly associated with SCN^−^ concentration (Figure 5iii).

Furthermore, the existence of •OH radicals in the experimental system was checked by the TA test. In general, the fluorescence signal of 2-Hydroxy-1,4-benzenedicarboxylic acid shows up when the reaction between TA and •OH radicals generates 2-Hydroxy-1,4-benzenedicarboxylic acid, and the emission peak of fluorescence is usually located at around 424 nm (Figure 6A). Additionally, the antioxidant property of the citric and ascorbic acid was utilized to monitor the presence of •OH radicals on the surface of SCN^−^-capped Au QDs. As presented in Figure 6B, the absorbance intensity of the color formed by SCN^−^-capped Au QDs decreased by approximately 50% and 49% in the presence of citric and ascorbic acid, respectively. This that those two antioxidants scavenged •OH radicals from the surface of SCN^−^-capped Au QDs. Hence, the huge amount of •OH radicals that scavenged SCN^−^-capped Au QDs helps to improve the detection sensitivity in this study.

### 3.5. Selectivity of the Present Study

The selectivity of the present nanozyme-based SCN^−^ assay was examined in the presence of BSA, fructose, sucrose, citrate, lysine, Mg^2+^, Ca^2+^, EDTA, SO_4_^2−^, CO_3_^2−^, Cl^−^, CH_3_CO^−^ and NO_3_^−^. As shown in Figure 7A, SCN^−^ showed a significantly higher absorbance intensity in comparison with the other interfering substances, confirming that the proposed colorimetric method has a much higher selectivity towards SCN^−^.

### 3.6. The Validation of the Proposed Assay in Spiked Milk Samples

The feasibility of the present SCN^−^ detection method was assessed by use in spiked milk samples. The findings reveal that the absorbance intensity was proportional to the SCN^−^ concentration, and the calculated LOD was 15 nM (Figure 7B). Moreover, the validation of this method was performed by checking the percentage of recovery in commercial milk. The results show the approximately 99–105% recovery of SCN^−^ with a relative standard deviation (RSD) of <3% (Table 1). This firmly establishes that the present method is highly reliable and accurate. In general, the thiocyanate contents in individual milk samples ranged from 6.01 to 8.92 mg/L, with an average of 7.30 ± 0.13. Hence, the proposed method can be used to measure the thiocyanate content in the milk.

### 3.7. Comparison Analysis

A comparison study of the analytical performance in SCN^−^ detection with other reports was undertaken, and the outcome is presented in Table 2. As seen in this table, the methodology proposed in this study allows for a very low LOD (8 nM) compared to the other reported articles.

## 4. Conclusions

In this study, a unique nanozymatic SCN^−^ group detection strategy has been reported based on the special interaction of the thiol group with the gold surface and the •OH radical scavenging activity of the SCN^−^. Herein, a new preparation method of positively charged Au QDs was developed, which showed its nanozymatic nature in the existence of SCN^−^, H_2_O_2_ and TMB. Then, a quantification of SCN^−^ was formulated by monitoring the change in the developed color, and the calculated LOD value was 8 nM and 15 nM in PBS and milk samples, respectively. The present nanozyme-based SCN^−^ sensing method was very specific to the target analyte, and 99–105% recovery was found in the complex matrix, i.e., the commercial milk sample. Moreover, the sensitivity of the present sensing strategy compared favorably to the others reported in the articles. Hence, this proposed method will provide a different sensing approach for colorimetric SCN^−^ detection, which can be easily translated into a real-life application.

## Figures and Tables

**Figure 1 foods-11-01189-f001:**
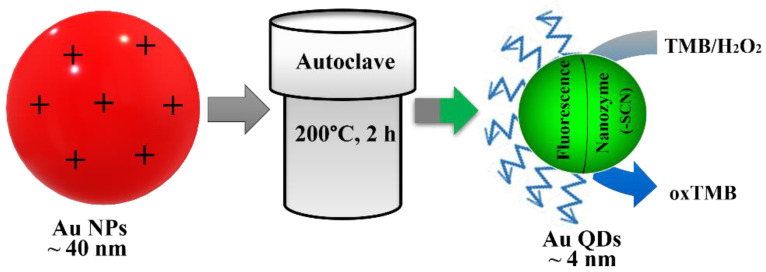
Schematic illustration of the present SCN^−^ detection.

**Figure 2 foods-11-01189-f002:**
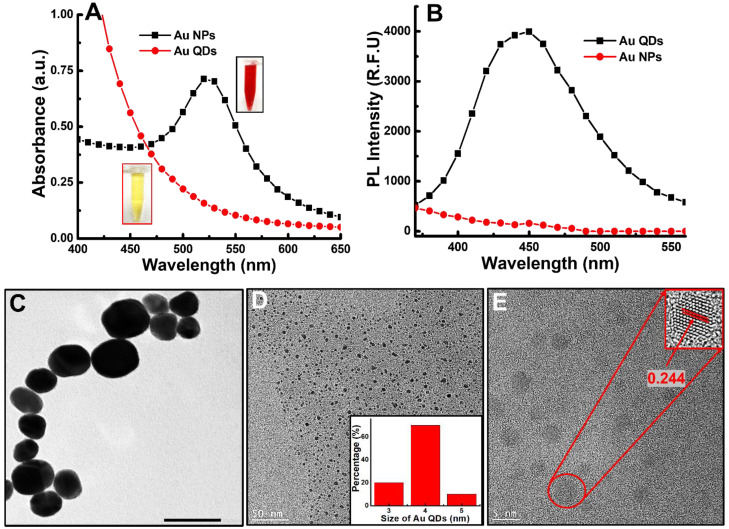
Identification of Au NPs and Au QDs: (**A**) absorbance of Au NPs and Au QDs; (**B**) fluorescence spectra of Au NPs and Au QDs; (**C**) TEM image of Au NPs: (**D**) HR-TEM image of Au QDs (inset: size distribution); (**E**) close view of Au QDs (inset: lattice fringes of Au QDs).

**Figure 3 foods-11-01189-f003:**
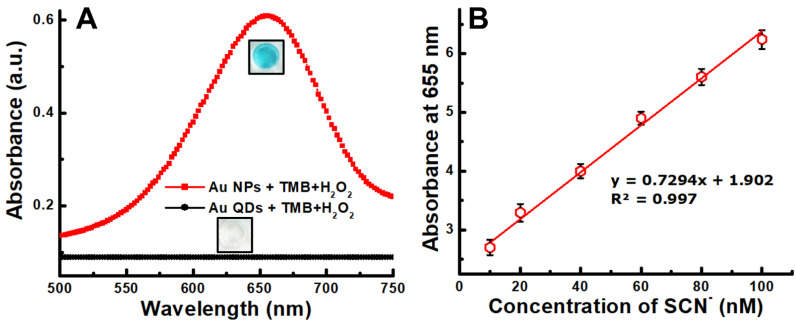
Nanozymatic activity of Au NPs and Au QDs (**A**) (inset: the color of the solution) and (**B**) calibration plot of differently concentrated SCN^−^ (absorbance at 655 nm).

**Figure 4 foods-11-01189-f004:**
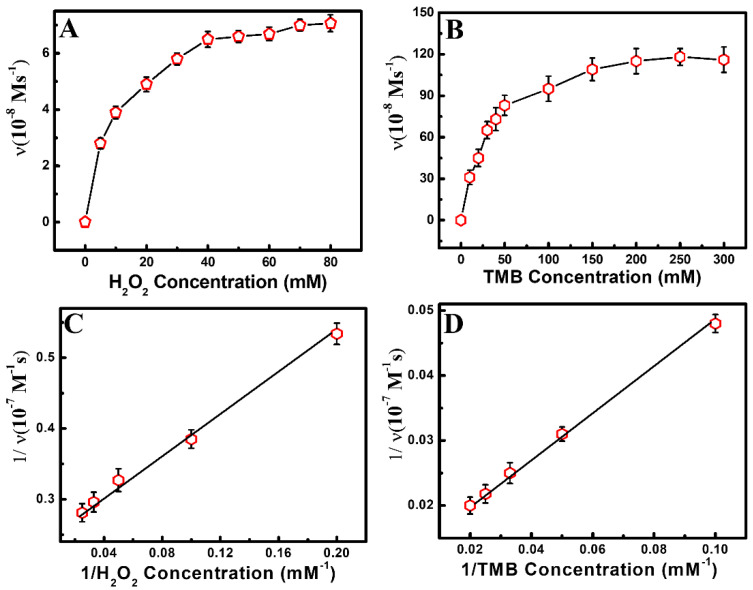
Kinetic study of Au QDs: (**A**,**C**) 5 mM TMB with various concentrations of H_2_O_2_, (**B**,**D**) 10 mM H_2_O_2_ with various concentrations of TMB.

**Figure 5 foods-11-01189-f005:**
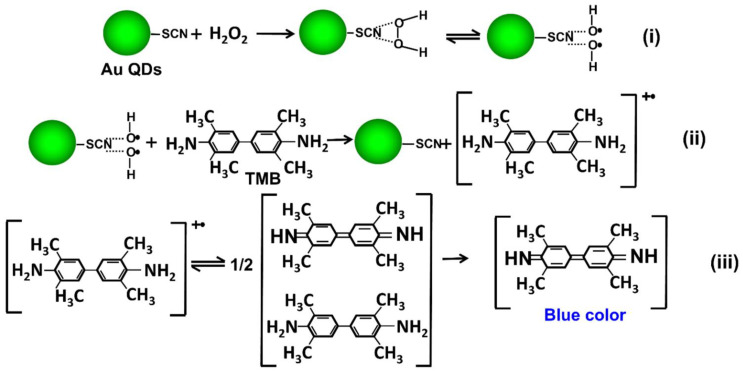
Mechanism study of Au QDs’ enzyme-like nature: (**i**) formation and adsorption of •OH radicals on the surface of Au QDs; (**ii**) oxidation reaction of TMB initiated by •OH radicals and (**iii**) formation of a dimer structure of oxidized TMB molecules that turns the solution color to deep blue.

**Figure 6 foods-11-01189-f006:**
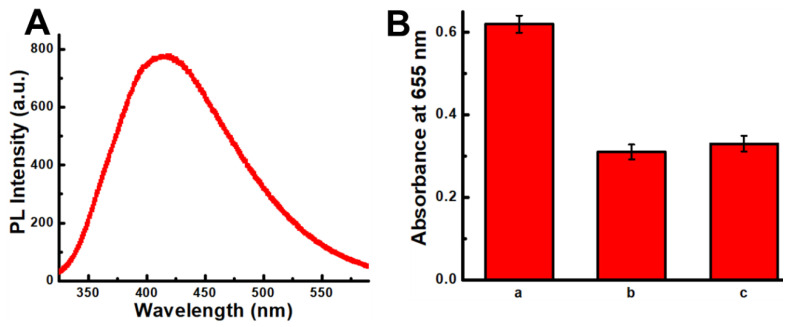
(**A**) Fluorescence emission of 2-Hydroxy-1,4-benzenedicarboxylic acid. (**B**) The absorbance intensity of SCN^−^-Au QDs in the presence of H_2_O_2_ and TMB (a); after adding citric acid (b) and ascorbic acid on sample a, separately (c).

**Figure 7 foods-11-01189-f007:**
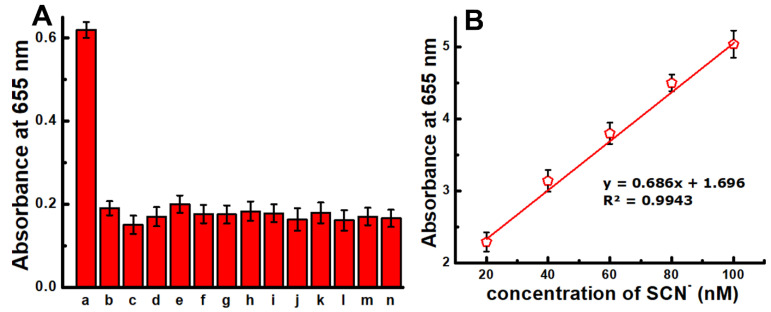
The specificity and SCN^−^ detection in spiked milk samples: (**A**) specificity of the proposed assay (a: SCN^−^, b: sucrose, c: citrate, d: BSA, e: fructose, f: SO_4_^2−^, g: Mg^2+^, h: Ca^2+^, i: EDTA, j: *l*-lysine, k: CO_3_^2−^, l: NO_3_^−^, m: CH_3_CO^−^, *n*: Cl^−^); (**B**) SCN^−^ concentration vs. absorbance intensity curve measured in the spiked milk samples.

**Table 1 foods-11-01189-t001:** Precision (RSD%) and percentage of recovery in the present study.

SCN^−^ Added (nM)	SCN^−^ Found (nM)	RSD (%)	Recovery (%)
20	19.8	1.5	99.00
40	42.18	2.1	105.45
60	61.90	2.4	103.16
80	79.10	2.9	98.87
100	103.12	1.7	103.12

**Table 2 foods-11-01189-t002:** Comparative analysis of the proposed method.

Nanomaterials	Techniques	LOD (nM)	Ref. No
CTAB@Au NPs	Colorimetric	100	[34]
Citrate@Au NPs	Colorimetric	36	[35]
Tween20@Au NPs	Colorimetric	200	[36]
Citrate@Au NPs	Colorimetric	140	[37]
Cysteamine@Au NPs	Colorimetric	200	[38]
C-dots@Au NPs	Optical	160	[39]
AuNPs@Ag NPs	SERS	50	[40]
Au QDs	Colorimetric	8	Present work

## Data Availability

The data presented in this study are available on request from the corresponding author.

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
