# Peer review of "Positively Charged Gold Quantum Dots: An Nanozymatic “Off-On” Sensor for Thiocyanate Detection"

_foods, 2022, doi:10.3390/foods11091189_

Round 1
Reviewer 1 Report
Manuscript No: 1658600
[1] Particularly, the biosensing application of nanomaterials in the food sector through the colorimetric method, electrochemical method, fluorescence method or surface-enhanced Raman scattering technique has gained numerous research interests because of effortless preparation, stability, bioconjugation and improved sensitivity.
Please provide suitable latest references for the above-mentioned techniques.
[2] Line 92- What is CH3CONa? Line 177- change color to colorless.
[3] Why not the pH of PBS buffer is mentioned in the experiment?
[4] As TMB is colourless in step 2 and authors claim the blue color is due to upper structure containing imine bond in the complex shown in square bracket. Then why the complex contains TMB in addition to imine derivative. It will affect the sensitivity of the nanozyme. Please explain.
[5] At what pH and temp the reaction has been carried out.
[6] What is the nanozymatic rate at different SCN- concentration? Explain it by carrying out Michelis-Menten equation.
[7] What is the reason to select AuQD? Which other QDs were used for comparison?
Author Response
Reviewer #1:
- Particularly, the biosensing application of nanomaterials in the food sector through the colorimetric method, electrochemical method, fluorescence method or surface-enhanced Raman scattering technique has gained numerous research interests because of effortless preparation, stability, bioconjugation and improved sensitivity.
Please provide suitable latest references for the above-mentioned techniques.
The authors respect the Reviewer's opinion. To address this comment, the authors have added the suitable reference in the introduction part and made several other modifications in the revised manuscript. We hope that the reviewer finds the manuscript significantly improved and now considers it suitable for publication in Foods Journal.
- Line 92- What is CH3CONa? Line 177- change color to colorless.
The authors appreciate the comment. It should be “CH3COONa” instead of “CH3CONa”. We have now corrected it in the revised manuscript for more clarification (Line 100).
- Why not the pH of PBS buffer is mentioned in the experiment?
The authors appreciate the comment. We have now mentioned the pH value in the revised manuscript for more clarification (Line 115).
- As TMB is colourless in step 2 and authors claim the blue color is due to upper structure containing imine bond in the complex shown in square bracket. Then why the complex contains TMB in addition to imine derivative. It will affect the sensitivity of the nanozyme. Please explain.
We highly appreciate reviewers’ constructive comments. We have corrected the Figure 5 (Fig. 4 in the old version) in the revised version.
- At what pH and temp the reaction has been carried out.
The reactions were carried out at pH 7.5 under room temperature condition. We have now mentioned the pH and temperature value in the revised manuscript for more clarification (Line 115 & 117).
- What is the nanozymatic rate at different SCN- concentration? Explain it by carrying out Michelis-Menten equation.
Highly appreciated the comments. Based on the reviewers’ comments, we have performed the kinetic study of SCN-capped Au QDs (Figure 4). For your convenience, the newly added sentences are given below:
3.4 Kinetic study on the nanozymatic activity of Au QDs
The apparent steady-state kinetic study was utilized to investigate the reaction kinetic parameters of SCN- capped Au QDs. At first, the non-enzymatic reaction of SCN- capped Au QDs was performed at different concentrations of H2O2 and TMB with variable time. Then, Michaelis-Menten equation (Fig. 4 A&B) and Lineweaver-Burk double reciprocal plot (Fig. 4 C&D) was employed to get the reaction kinetic parameters. The results of kinetic study revealed that the Au QDs had a lower Michaelis-Menten constant (Km) value (0.38) towards TMB compared to Horseradish Peroxidase (HRP) (0.43). Moreover, as-synthesized Au QDs had a lower Km value (0.34) towards H2O2 than HRP (3.7). Those finding indicates the stronger affinity of Au QDs towards H2O2 and TMB. The catalytic rate constant (Kcat) of Au QDs (5.8.00×103 s−1) was higher than HRP (4.00×103 s−1), indicating the non-enzymatic reaction of Au QDs was kinetically faster than natural enzyme (HRP)[34].
[7] What is the reason to select Au QD? Which other QDs were used for comparison?
Highly appreciated the comments. We have explained the reason to select Au QDs in the last paragraph of the Introduction part. For your convenience, the sentences are given below:
“It is well-known that positively charged Au NPs showed better nanozymatic activity compared to negatively charged Au NPs because of the strong electrostatic interaction with •OH radical. Moreover, the smaller-sized nanomaterials performed better nanozymatic activity compared to their bulk part due to the large surface-to-volume ratio that allows more substrate to interact with it. Considering all these parameters, the synthesis of positively charged Au QDs might open a new era in nanozymatic biosensing applications.”
For comparison study, we have used some recent reported nanomaterials-based thiocyanate detection articles (Table 2). Carbon quantum dots for thiocyanate detection has been compared with the present study.

Reviewer 2 Report
This manuscript reported a positively charged gold quantum dots sensor for thiocyanate detection. Here gives some suggestions for improvement of the manuscript:
- The abstract section needs to be revised, more informationshould be mentioned, such as the LOD value and the percentage of recovery.
- “Introduction” parts should be modified. The authors should introduce previous common methods of measuring thiocyanate and compare the innovations of this work with previous use of measurement strategies.
- Please define some abbreviations at it is first use in the text. Such as, “Au QDs”(line 81), “TMB”(line 109).
- “NIS”(line 24), “MPO” (line 32) and “o” (line 219) should be removed, and please check the whole manuscript carefully.
- More information of instrument, reagents and materials used in this study should be provided.
- In Fig.1, “TMBZ”should be changed to “TMB”, “oxTMBZ” should be changed to “oxTMB”.
- In Fig.4, the role of thiocyanate should be highlighted.
- The kinetic parameter of Au QDs should be studied, such as Michaelis constant.
- The information of milk sample needs to be reconfirmed, such as “raw-milk samples”(line 20) or “commercial milk” (line 210),
- What is the international standard for thiocyanatecontent in milk samples? Can it be measured by the established method?
- The quality of all figures needed to be greatly improved.
Author Response
Reviewer #2: This manuscript reported a positively charged gold quantum dots sensor for thiocyanate detection. Here gives some suggestions for improvement of the manuscript:
- The abstract section needs to be revised, more information should be mentioned, such as the LOD value and the percentage of recovery.
The authors respect the Reviewer's opinion. To address this comment, the authors have rewritten the abstract part, and made several other modifications in the revised manuscript. We hope that the reviewer finds the manuscript significantly improved and now considers it suitable for publication in Foods Journal.
- “Introduction” parts should be modified. The authors should introduce previous common methods of measuring thiocyanate and compare the innovations of this work with previous use of measurement strategies.
The authors respect the Reviewer's opinion. To address this comment, the authors have introduced the common detection technique of thiocyanate in the introduction part. For your convenience, the sentences are given below:
Numerous SCN- detection methods have been proposed over the last few years, for example, traditional colorimetric assay, chromatographic method, fluorescence method, electrophoresis method, chromatography technique, electrochemical method and SERS technique[8-11]. Unfortunately, those techniques need a longer time, more chemicals, and are costly. Hence, a sensitive, reliable, and cheaper SCN- detection technique is required.
- Please define some abbreviations at it is first use in the text. Such as, “Au QDs”(line 81), “TMB”(line 109).
The authors respect the Reviewer's opinion. We have mentioned the abbreviations in the abstract part.
- “NIS”(line 24), “MPO” (line 32) and “o” (line 219) should be removed, and please check the whole manuscript carefully.
Thank you for your nice comments. All the suggested changes and several other modifications has been made in the revised manuscript.
- More information of instrument, reagents and materials used in this study should be provided.
We have expanded the list of reagents and materials in the revised version.
- In Fig.1, “TMBZ”should be changed to “TMB”, “oxTMBZ” should be changed to “oxTMB”.
To address this comment, we have modified Fig 1 in the revised version.
- In Fig.4, the role of thiocyanate should be highlighted.
Based on the reviewer’s suggestion, we have modified Fig 4 in the revised version.
- The kinetic parameter of Au QDs should be studied, such as Michaelis constant.
Highly appreciated the comments. Based on the reviewers’ comments, we have performed the kinetic study of SCN-capped Au QDs (Figure 4). For your convenience, the newly added sentences are given below:
3.4 Kinetic study on the nanozymatic activity of Au QDs
The apparent steady-state kinetic study was utilized to investigate the reaction kinetic parameters of SCN- capped Au QDs. At first, the non-enzymatic reaction of SCN- capped Au QDs was performed at different concentrations of H2O2 and TMB with variable time. Then, Michaelis-Menten equation (Fig. 4 A&B) and Lineweaver-Burk double reciprocal plot (Fig. 4 C&D) was employed to get the reaction kinetic parameters. The results of kinetic study revealed that the Au QDs had a lower Michaelis-Menten constant (Km) value (0.38) towards TMB compared to Horseradish Peroxidase (HRP) (0.43). Moreover, as-synthesized Au QDs had a lower Km value (0.34) towards H2O2 than HRP (3.7). Those finding indicates the stronger affinity of Au QDs towards H2O2 and TMB. The catalytic rate constant (Kcat) of Au QDs (5.8.00×103 s−1) was higher than HRP (4.00×103 s−1), indicating the non-enzymatic reaction of Au QDs was kinetically faster than natural enzyme (HRP)[34].
- The information of milk sample needs to be reconfirmed, such as “raw-milk samples”(line 20) or “commercial milk” (line 210),
To address this comment, the authors have amended the sentence in the revised version (Line 20).
- What is the international standard for thiocyanate content in milk samples? Can it be measured by the established method?
The thiocyanate contents in individual milk samples ranged from 6.01-8.92 mg/litre with an average of 7.30 ± 0.13. Hence, the proposed method can be measured the thiocyanate content in the milk.
- The quality of all figures needed to be greatly improved.
To address this comment, the authors have redrawn and improved the quality of all the figures in the revised version.

Round 2
Reviewer 2 Report
The present manuscript have gained significant improvement after revision. Here gives some suggestions for further improvement of the manuscript:
- The instruments used for characterization of Au NPs and Au QDs should be provided.
- Lines 179-180, the reference should be provided.
- Line 205-208, “Fig. 4”should be changed to “Fig. 5”.
- The international standard for thiocyanate content in milk samples should be added in the “3.6 The validation of the proposed assay in spiked milk samples”section.
- The reaction kinetic parameters of SCN-capped Au QDs should be mentioned in Abstract section.
Author Response
Reviewer #2: The present manuscript have gained significant improvement after revision. Here gives some suggestions for further improvement of the manuscript:
- The instruments used for characterization of Au NPs and Au QDs should be provided.
The authors respect the Reviewer's opinion. To address this comment, the authors has added all the information of instruments used for the characterization of Au NPs and Au QDs in the revised version. For your convenience, the sentences are given below:
The color intensity was monitored using a UV-visible spectrometer (Synergy H1, BioTek, USA).
The fluorescence signal was registered by spectrophotometer (Synergy H1, BioTek, USA).
The microscopic study of Au NPs and Au QDs was performed using FEI Titan 80—300 LB (USA).
The fluorescence lifetime of Au QDs was measured using DeltaFlex TCSPC Lifetime Fluorometer (Horiba, London, Canada)
- Lines 179-180, the reference should be provided.
The authors have provided the reference in the revised version.
- Line 205-208, “Fig. 4”should be changed to “Fig. 5”.
The author has changed the Figure number as suggested by the reviewer.
- The international standard for thiocyanate content in milk samples should be added in the “3.6 The validation of the proposed assay in spiked milk samples”section.
The authors respect the Reviewer's opinion. To address this comment, the authors have introduced the sentences in the section “3.6 The validation of the proposed assay spiked milk samples”.
- The reaction kinetic parameters of SCN-capped Au QDs should be mentioned in Abstract section.
To address this comment, the authors have mentioned the reaction kinetic parameters of Au QDs in the abstract section.
